# Profiling Antibiotic Resistance in *Acinetobacter calcoaceticus*

**DOI:** 10.3390/antibiotics11070978

**Published:** 2022-07-20

**Authors:** Janiece S. Glover, Taylor D. Ticer, Melinda A. Engevik

**Affiliations:** 1Department of Regenerative Medicine and Cell Biology, Medical University of South Carolina, Charleston, SC 29425, USA; gloverja@musc.edu; 2Department of Microbiology and Immunology, Medical University of South Carolina, Charleston, SC 29425, USA; ticer@musc.edu

**Keywords:** *Acinetobacter*, *Acinetobacter calcoaceticus*, antibiotic resistance

## Abstract

Background: *Acinetobacter* spp. have emerged as troublesome pathogens due to their multi-drug resistance. The majority of the work to date has focused on the antibiotic resistance profile of *Acinetobacter baumannii*. Although *A.* *calcoaceticus* strains are isolated in the hospital setting, limited information is available on these closely related species. Methods & Results: The computational analysis of antibiotic resistance genes in 1441 *Acinetobacter* genomes revealed that *A. calcoaceticus* harbored a similar repertoire of multi-drug efflux pump and beta-lactam resistance genes as *A. baumannii*, leading us to speculate that *A. calcoaceticus* would have a similar antibiotic resistance profile to *A. baumannii*. To profile the resistance patterns of *A. calcoaceticus*, strains were examined by Kirby–Bauer disk diffusion and phenotypic microarrays. We found that *Acinetobacter* strains were moderately to highly resistant to certain antibiotics within fluoroquinolones, aminoglycosides, tetracyclines, and other antibiotic classes. These data indicate that *A. calcoaceticus* has a similar antibiotic resistance profile as *A. baumannii* ATCC 19606. We also identified that all *Acinetobacter* species were sensitive to 5-fluoroorotic acid, novobiocin, and benzethonium chloride. Conclusion: Collectively, these data provide new insights into the antibiotic resistance in *A. calcoaceticus* and identify several antibiotics that could be beneficial in treating *Acinetobacter* infections.

## 1. Introduction

*Acinetobacter* spp. are Gram-negative bacteria that have become a pressing health concern over the past few decades. Worldwide, *Acinetobacter* spp. make up 20% of the infections seen in the hospital setting and are considered the main cause of hospital-acquired bacteremia and pneumonia [1]. Once infection occurs, professionals are challenged with formulating strategic treatment plans for patients. This is because *Acinetobacter* spp. are characterized by their intrinsic antimicrobial resistance [2]. *Acinetobacter* strains are increasingly resistant to the major classes of antimicrobials, including beta-lactams, aminoglycosides, and fluoroquinolones [3]. Due to the high degree of resistance to multiple classes of antimicrobials, these microbes are classified as “multi-drug resistant” or “MDR” [2,4]. Unfortunately, the prevalence of MDR *Acinetobacter* isolates has increased in recent years [3], and, according to the World Health Organization (WHO), *Acinetobacter* are among the most serious MDR organisms [5].

The *Acinetobacter baumannii-calcoaceticus* (ABC) complex is a group of phenotypically indistinguishable opportunistic pathogens that cause hospital-acquired bacteremia and pneumonia. Among the ABC complex members, *A. baumannii, A. pittii*, and *A. nosocomialis* are the most commonly isolated strains [6,7]. *A. calcoaceticus* strains are also isolated, but less frequently [8]. Based on the frequency of isolation in infections, the majority of studies examining the antimicrobial resistance in *Acinetobacter* have focused on *A. baumannii* [2]. These studies have found that *A. baumannii* strains are resistant to most first-line antibiotics. Several studies have examined the antibiotic resistance in ABC complex isolates but have not distinguished between species [9,10]. Determining the antibiotic profile of strains within ABC complexes is important because some organisms may be more resistant than others. By characterizing the antibiotic profiles of other strains, such as *A. calcoaceticus*, we may be able to identify antibiotics to which multiple species are sensitive. Currently, few studies have specifically examined the resistance profile of *A. calcoaceticus* strains. Herein, we sought to address this gap in knowledge by examining the antibiotic sensitivity profile of commercially available *A. calcoaceticus* and clinical isolates of *A. calcoaceticus*.

## 2. Results

Despite the diverse targets of antibiotic classes (Figure 1A), *Acinetobacter* species are well documented for their ability to counteract the effects of antibiotics through multidrug efflux pumps and targeted mechanisms. Since antibiotic resistance mechanisms are commonly studied in *A. baumannii* species, we sought to compare the genomic capacity of *A. calcoaceticus* strains to resist antibiotics to that of *A. baumannii* and unclassified *Acinetobacter* strains. We identified 1290 genomes of *A. baumannii*, 23 genomes of *A. calcoaceticus*, and 128 genomes of unclassified *Acinetobacter* spp. in the Integrated Microbial Genomes (IMG) database (img.jgi.doe.gov; accessed on 9 October 2021) and examined key genes in the antibiotic resistance pathways, as identified in the KEGG database (KEGG: Kyoto Encyclopedia of Genes and Genomes, https://www.genome.jp; accessed on 9 October 2021) (Figure 1B). We found several multidrug efflux pump genes, including *TolC/AcrA/AcrB, MexA/MexB/OprM*, *OMP/RND/MFP*, and *AdeA/AdeB/AdeC*, in the majority of examined *Acinetobacter* genomes. These efflux proteins are known to secrete various antibiotics out of the outer cell wall, thereby limiting the effectiveness of these antimicrobial compounds. *A. baumannii* is well characterized for its resistance to beta-lactams via alterations in outer membrane proteins (OMPs) production, beta-lactamases, penicillin-binding proteins (PBPs), and the increased activity of efflux pumps [11]. In our genome analysis, we found that *A. baumannii*, *A. calcoaceticus*, and *Acinetobacter* spp. genomes did not have genes encoding the beta-lactam transporters *OprD*, *OmpF*, *OmpC*, *OmpU*, or *PorB*, indicating that these bacteria can limit the uptake of beta-lactams into the outer membrane. We also found that >95% of the *A. baumannii* and *A. calcoaceticus* genomes harbored Class C (*AmpC*) beta-lactamases. Some *A. baumannii* genomes contained the genes for Class A (*blaTEM*; 20.5%), Class D (*pxa23*; 22.9%), and Class B (*bla2*; 3.4%) beta-lactamases. Only a few *A. calcoaceticus* genomes had Class A (4.3%) beta-lactamases, and no genomes had Class B or D beta-lactamases. Interestingly, only 21% of unclassified *Acinetobacter* species harbored Class C beta-lactamases, and 8.6% had Class C beta-lactamases. These findings suggest that *A. calcoaceticus* is similar to *A. baumannii* in terms of its genomic capacity to resist antibiotics.

To examine the antibiotic resistance profile of *A. calcoaceticus*, we examined commercially available *A. calcoaceticus* ATCC 23055 and *A. calcoaceticus* CB1 and *A. calcoaceticus* clinical isolates (M31602, T82482, X75393, and M53152) using the Kirby Bauer Disc Diffusion Assay (Table 1). For comparison, we also examined *Acinetobacter* spp. ATCC 27244 and *A. baumannii* ATCC 19606. We observed that all *Acinetobacter* strains were resistant to erythromycin (macrolide) and colistin (miscellaneous antibiotic). All strains had an intermediate resistance to ceftazidime (cephalosporin), meropenem (carbapenem), and tetracycline (tetracycline) and were sensitive to imipenem (carbapenem), amikacin (aminoglycoside), gentamicin (aminoglycoside), tobramycin (aminoglycoside), and tigecycline (tetracycline) (Table 1).

To expand our antibiotic profile, we also employed Biolog phenotypic microarrays, which contain a variety of antibiotics. The growth with antibiotics was contrasted to the growth without antibiotics. We considered bacteria to be resistant if >75% growth was observed in the presence of antibiotics, moderately resistant if the growth was between 25% and 75% with antibiotics, and sensitive if the bacteria grew <25% with antibiotics. We first examined the cephalosporins cefazolin, ceftriaxone, and cephalothin as well as the carbapenem imipenem (Figure 2A). All *Acinetobacter* strains were moderately resistant to the cephalosporins and sensitive to the carbapenem imipenem. We also examined fluoroquinolones and found that all the strains were moderately resistant to the antibiotics lomefloxacin, enoxacin, and ofloxacin (Figure 2B). We then examined aminoglycosides (Figure 2C). All strains were moderately resistant or resistant to kanamycin, neomycin, capreomycin, gentamicin, colistin, and amikacin (Figure 2C). Notably, *A. baumannii* ATCC 19606 was more resistant than the *A. calcoaceticus* strains to paromomycin, sisomicin, tobramycin, and spectinomycin.

*A. calcoaceticus* strains, unclassified *Acinetobacter*, and *A. baumannii* were moderately resistant to chlortetracycline and demeclocycline (Figure 3A). All *Acinetobacter* strains were sensitive to tetracycline, minocycline, and penimepicycline (Figure 3A). These data indicate that *A. calcoaceticus* has a similar resistance profile to tetracyclines as *A. baumannii*. We also examined beta-lactam antibiotics and observed a strain-dependent and antibiotic-specific resistance (Figure 3B). We found that all *Acinetobacter* strains were moderately resistant to penicillin G, carbenicillin, cloxacillin, nafcillin, and amoxicillin. Oxacillin was more effective in limiting the growth of the *Acinetobacter* strains compared to the other beta-lactams tested (Figure 3B). These data emphasize that, even within classes such as beta-lactams, resistance is specific to the individual antibiotic.

Finally, we examined antibiotics that fall into the miscellaneous category. Significant differences in antibiotic resistance between *A. calcoaceticus* and *A. baumannii* were observed in response to sulfonamides: sulfadiazine sulfathiazole and sulfa-methoxazole (Figure 3C). We found that all *A. calcoaceticus* strains were sensitive to these antibiotics, while *A. baumannii* was more resistant. In response to acids, *A. baumannii* had a higher degree of resistance to D,L-serine hydroxamate than the *A. calcoaceticus* and unclassified *Acinetobacter* strains (Figure 3C). Surprisingly, all the *Acinetobacter* strains were sensitive to the acid-derived antibiotic 5-fluoroorotic acid, rifampicin, dodecyltrimethyl ammonium bromide, and novobiocin (Figure 3C), suggesting that these antibiotics may be useful in treating *Acinetobacter* species infections.

## 3. Discussion

The limitations surrounding the treatment for *Acinetobacter* infections continue to be a healthcare concern, and while multiple studies have identified antibiotic resistance in *A. baumannii* strains, there is only a partial characterization of the antibiotic-resistance in *A. calcoaceticus* infections. Based on our computation analysis, the genome of *A. calcoaceticus* resembles that of *A. baumannii* in terms of several antibiotic resistance genes. Consistent with this notion, we found that *A. baumannii* had a similar sensitivity profile to *A. calcoaceticus* for most antibiotics. For example, *A. calcoaceticus* and *A. baumannii* were both resistant to carbenicillin, cefazolin, ceftriaxone, and cephalothin and sensitive to 5-fluoroorotic acid, benzethonium chloride, and novobiocin. However, we did note some differences. These data suggest that although *A. calcoaceticus* is antibiotic-resistant, *A. baumannii* likely has a larger antibiotic resistant repertoire. Together, these data provide insight into the antibiotic sensitivity of *A. calcoaceticus* strains and highlight that antibiotic resistance is strain-dependent and antibiotic-specific.

To date, there have only been a few studies that directly focus on *A. calcoaceticus*. One study found that one strain of *A. calcoaceticus* was resistant to ampicillin, clindamycin, chloramphenicol, and cefazolin [12]. To the best of our knowledge, this is the only study to specifically examine antibiotic resistance in *A. calcoaceticus*, although many studies have examined *A. baumannii-calcoaceticus* (ABC) complexes where the specific species cannot be distinguished. In our study, we found several antibiotics that significantly reduced the growth of *Acinetobacter* spp. in drug classifications that are not normally used to treat these organisms. For example, we found that *Acinetobacter* spp. are sensitive to novobiocin. Novobiocin interferes with the ATPase activity of DNA gyrase and inhibits the DNA repair/replication processes initiated by double-strand breaks [13]. Importantly, novobiocin has been shown to inhibit the acquisition of antimicrobial resistance through DNA damage-induced mutagenesis in the *A. baumannii* strain ATCC 17978 [13] and be effective against *A. baumannii* AB5075 [14]. Our study demonstrates that novobiocin is also effective against *A. calcoaceticus* strains and an unclassified *Acinetobacter* species. These data suggest that novobiocin could serve as a potential therapy for patients with *Acinetobacter* infections. Additionally, combination drugs can be explored, such as novobiocin and rifampicin. Recently, nine dilipid ultrashort tetrabasic peptidomimetics (dUSTBPs) were found to potentiate novobiocin and rifampicin activity against multidrug-resistant (MDR) clinical isolates of *A. baumannii* [15].

We also found that all *Acinetobacter* strains were highly sensitive to 5-fluoroorotic acid. 5-fluoroorotic acid has been used experimentally for many years in yeasts and archaea studies [16]. Non-toxic 5-fluoroorotic acid is converted to the toxic 5-fluoro-UMP by the two genes encoding orotate phosphoribosyltransferase (pyrE) and orotidine 5-phosphate decarboxylase (pyrF) in micro-organisms. Our data indicate that *A. calcoaceticus,* unclassified *Acinetobacter* spp. ATCC 27244, and *A. baumannii* ATCC 19606 are highly sensitive to 5-fluoroorotic acid. This finding is consistent with a previous study that found that the *A. baumannii* strains DR2 and AB067 were also sensitive to 5-fluoroorotic acid [17]. Importantly, human cell lines (HT-1080, IMR-90, HeLa S3, and HL-60) and a mouse cell line (L-1210) have been shown to be relatively tolerant of 5-fluoroorotic acid [18], indicating that 5-fluoroorotic acid could be used to treat infections in patients.

A major limitation of our study is that we did not identify minimum inhibitor concentrations (MICs) for the strains. Additionally, our Biolog Phenotypic Microarrays were performed in a chemically defined medium. The advantage of using a chemically defined medium is that the supernatant is amenable to downstream LC-MS/MS analysis since it does not contain components such as peptone and yeast extract. We have used this defined medium to examine the growth of diverse microbes and measure bacterial metabolites and proteins by LC-MS/MS [19,20,21,22,23]. However, this medium has not been validated for antibiotic studies, and the current gold standard medium is Mueller Hinton broth. In the future, it will be important to obtain standard MICs with the *A. calcoaceticus* strains.

## 4. Conclusions

This work sheds light onto how *A. calcoaceticus* strains respond to antibiotics and demonstrates that *A. calcoaceticus* behaves in a similar manner as *A. baumannii* in terms of antibiotic resistance. We believe this work can provide new insights into *Acinetobacter* resistance and new targets for antibiotic therapy.

## 5. Materials and Methods

### 5.1. Bacterial Culture Conditions

*A.calcoaceticus* ATCC 23055 and *A. calcoaceticus* Carolina Biological (CB1) strains were purchased for use in this study. Four clinical isolates of *A. calcoaceticus* (M31602, T82482, X75393, and M53152) were obtained [24]. For comparison, *Acinetobacter baumannii* ATCC 19606 and *Acinetobacter* spp. ATCC 27244 were used. All strains were cultured on Leeds *Acinetobacter* agar, a selective and differential medium for *Acinetobacter* spp. [25] Single colonies were used to inoculate liquid cultures of brain–heart infusion (BHI). All bacteria were grown at 37 °C aerobically in 10 mL conical tubes and shaken overnight at 150 rpm. The bacterial growth of liquid cultures was assessed by optical density (OD_600nm_) on a Spectronic 200 spectrophotometer (ThermoFisher, Waltham MA, USA).

### 5.2. Antibiotic Treatment

The antibiotic sensitivity of *Acinetobacter* strains was determined using the Kirby–Bauer method. Antibiotic disks were purchased from FisherScientific (Waltham MA, USA) and include: 30 µg Amikacin (cat# CT0107B); 30 µg Ceftazidime (cat# CT0412B); 10 µg Gentamicin (cat# CT0024B); 10 µg Colistin (cat# CT0017B); 15 µg Erythocycline (cat# 237093); 10 µg Imipenem (cat# CT0455B); 10 µg Meropenem (cat# CT0774B); 30 µg Tetracycline (cat# 230998); 15 µg Tigecycline (cat# CT1841B); and 10 µg Tobramycin (cat# CT0056B). Bacteria were diluted at a McFarland of 0.5 and incubated on Mueller Hinton Agar plates overnight (20 h) at 37 °C with the antibiotic disks. Zone diameters were recorded, and the susceptibility results were categorized as sensitive, moderately resistant, or resistant based on the CLSI breakpoint criteria [26].

To measure the growth of *Acinetobacter* in the presence of antibiotics, overnight BHI cultures were adjusted to an OD_600nm_ = 0.1 in chemically defined media termed ZMB1 [27]. A total of 100 μL of this OD_600nm_ adjusted ZMB1 was placed into each 96 well of the PM11 and PM12B Phenotype Microarray Biolog plate (*n* = 4 per antibiotic; repeated two independent times). Growth was monitored after 16 h of incubation at 37 °C on a Synergy HT BioTek plate reader. Antibiotic-treated microbes were compared to untreated microbes in ZMB1, and strains were considered to be sensitive to an antibiotic when there was <50% growth with an applied antibiotic compared to untreated microbes. Strains were considered moderately resistant to an antibiotic when there was 25–50% growth and highly resistant to an antibiotic when there was >75% growth in antibiotics compared to growth without antibiotics.

### 5.3. Computational Analysis

*Acinetobacter* genomes were identified in the Integrated Microbial Genomes (IMG) database (img.jgi.doe.gov) available through the Joint Genomes Institute (JGI) (Version 6.0; Berkeley, CA, USA) [28]. We identified 1290 genomes of *A. baumannii*, 23 genomes of *A. calcoaceticus*, and 128 genomes of unclassified *Acinetobacter* spp. The genes involved in beta-lactam resistance were identified in the KEGG database (KEGG: Kyoto Encyclopedia of Genes and Genomes, https://www.genome.jp; accessed on 9 October 2021). Based on a KEGG pathway analysis, we queried the *Acinetobacter* genomes for the following antibiotic-resistant genes: *TolC* (K12340), *AcrA* (K03585), *AcrB* (K18138), *MexA* (K03585), *MexB* (K18138), *OrpM* (K18139), RND (K18146), MFP (K18145), OMP (K12340), *AdeA* (K18145), *AdeB* (K18146), *AdeC* (K18147), *blaTEM* (Class A; K18698), *Bla2* (Class B; K17837), *AmpC* (Class C; K20319), P*xa23* (Class D; K18793), *OprD* (K18093), *OmpF* (K09476), *OmpC* (K09475), *OmpU* (K08720), *PorB* (K18133), PBP1a/2 (K05366), PBP2 (K05515), and *Fts1* (K03587).

### 5.4. Statistics

The data are presented as the mean ± standard deviation. Comparisons between the groups were made with One-way Analysis of Variance (ANOVA). GraphPad was used to generate the graphs and statistics (GraphPad Software v9.3, Inc. La Jolla, CA, USA). A * *p* < 0.05 value was considered significant.

## Figures and Tables

**Figure 1 antibiotics-11-00978-f001:**
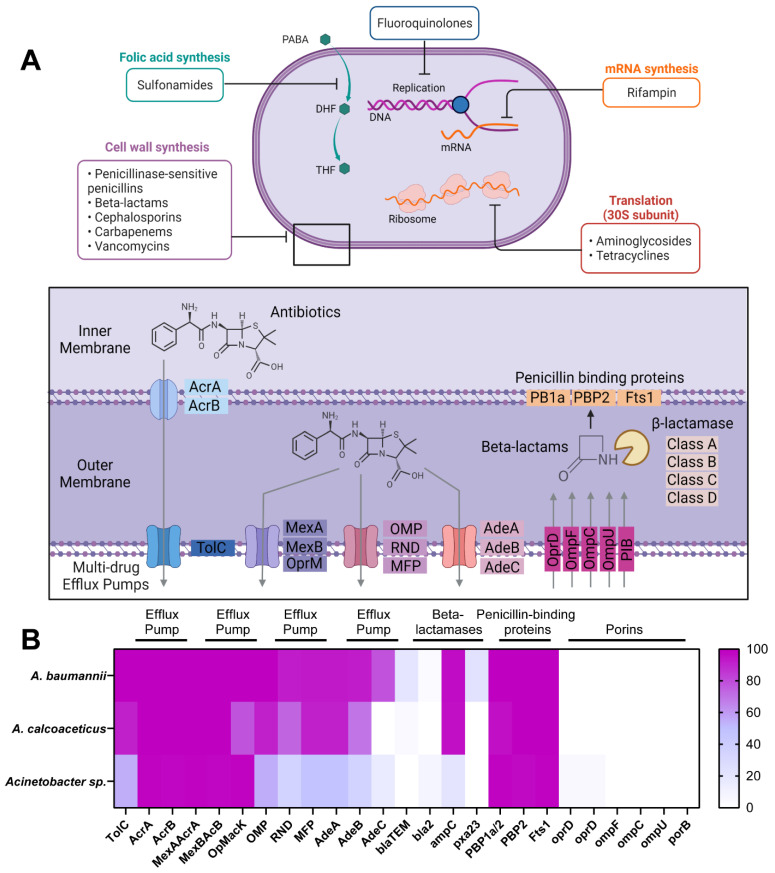
Genomic analysis of *Acinetobacter resistance* genes. (**A**) Diagram highlighting the mechanism of action of the antibiotics used in this study. The inset shows the specific mechanisms of antibiotic resistance previously identified in *Acinetobacter* species. (**B**) Heat map of the percentage of *A. baumannii*, *A. calcoaceticus,* or unclassified *Acinetobacter* spp. genomes that have at least one gene copy of antibiotic resistance genes.

**Figure 2 antibiotics-11-00978-f002:**
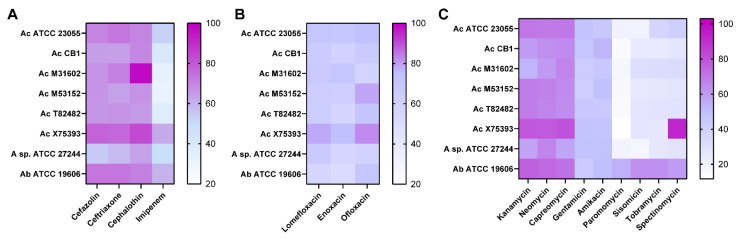
*A. calcoaceticus* resistance to cephalosporins, carbapenems, fluoroquinolones, and aminoglycosides. Antibiotic resistance was measured by the percentage (%) of growth at a 16 h time point with antibiotics compared to that without antibiotics. *A. calcoaceticus* ATCC 23055, *A. calcoaceticus* CB1; *A. calcoaceticus* M31602, *A. calcoaceticus* T82482, *A. calcoaceticus* X75393, *A. calcoaceticus* M53152, unclassified *Acinetobacter* ATCC 27244, and *A. baumannii* ATCC 19606 were grown with (**A**) cephalosporins, carbapenems, (**B**) fluoroquinolones, and (**C**) aminoglycosides.

**Figure 3 antibiotics-11-00978-f003:**
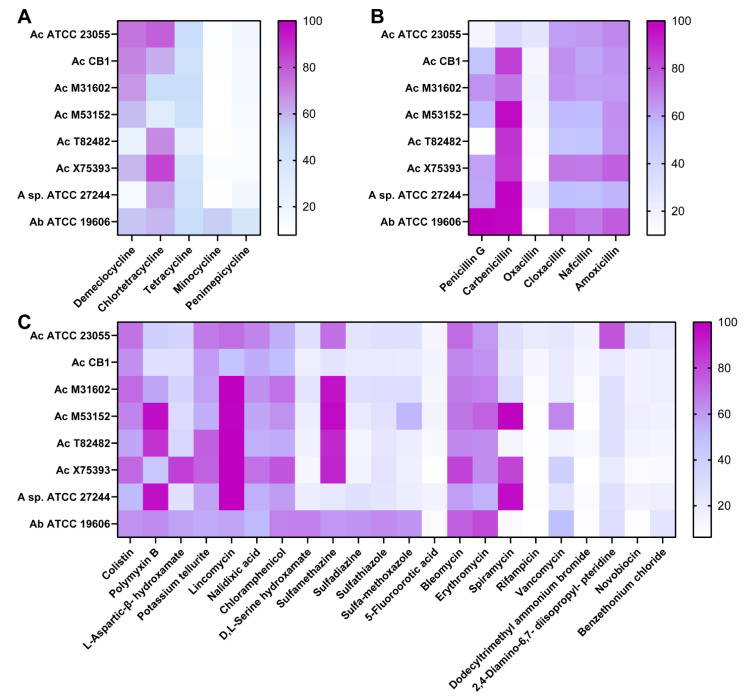
*A. calcoaceticus* resistance to tetracyclines, penicillin, and miscellaneous antibiotics. Antibiotic resistance was measured by the percentage (%) of growth at a 16 h time point with antibiotics compared to that without antibiotics. *A. calcoaceticus* ATCC 23055, *A. calcoaceticus* CB1, *A. calcoaceticus* M31602, *A. calcoaceticus* T82482, *A. calcoaceticus* X75393, *A. calcoaceticus* M53152, unclassified *Acinetobacter* ATCC 27244, and *A. baumannii* ATCC 19606 were grown with (**A**) tetracyclines, (**B**) penicillin, and (**C**) miscellaneous antibiotics.

**Table 1 antibiotics-11-00978-t001:** Measurements of the zones of inhibition of *Acinetobacter* strains (mm). R, resistant; I, intermediate resistant; S, sensitive.

				*A. calcoaceticus*						*A. baumannii*		*Acinetobacter* spp.	
Class	Antibiotics	Concentration	23055	CB1	M31602	M53152	T82482	X75393	Category	19606	Category	27244	Category
Cephalosporins	Ceftazidime	30 µg	19	20	19	19	19	19	I	19	I	19	I
Carbapenem	Imipenem	10 µg	28	30	29	29	29	29	S	29	S	29	S
Carbapenem	Meropenem	10 µg	25	24	24	24	24	24	I	24	I	24	I
Aminoglycoside	Amikacin	30 µg	20	21	20	20	20	20	S	20	S	20	S
Aminoglycoside	Gentamicin	10 µg	24	22	23	22	23	22	S	23	S	22	S
Aminoglycoside	Tobramycin	10 µg	21	21	21	21	21	21	S	21	S	21	S
Tetracyclines	Tetracycline	30 µg	15	16	15	15	15	15	I	15	I	15	I
Tetracyclines	Tigecycline	15 µg	23	24	23	23	23	23	S	23	S	23	S
Macrolides	Erythromycin	15 µg	15	11	13	12	13	12	R	12	R	12	R
Miscellaneous	Colistin Sulphate	10 µg	15	14	14	14	14	14	R	14	R	14	R

## Data Availability

All data are available upon request.

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
