# Peer review of "Profiling Antibiotic Resistance in Acinetobacter calcoaceticus"

_antibiotics, 2022, doi:10.3390/antibiotics11070978_

Round 1

Reviewer 1 Report

Authors presented a study entitled "Profiling antibiotic resistance in Acinetobacter calcoaceticus", they analized bioinformatically 1444 genomes publicly available of A. baumanii, A. coalcoaceticus and Acinetibacter spp.

The results are properly presented and discussed. I only suggest a brief review as follows:

Lines 56, 97, Figure 1 between lines 152-154 and Figure 8 between lines 304-306 are mentioned bacteria names that must be italicized.

Author Response

We appreciate the careful consideration of our manuscript by the reviewer and we thank the reviewer for identifying this oversight. We have now italicized the names of the bacteria.

Reviewer 2 Report

The Authors performed a study on profiling antibiotic resistance in Acinetobacter calcoaceticus. Despite being an interesting subject the used methodology and antimicrobial choice present several limitations:

- it is not clear che choice of antimicrobials. Most of the drugs are not used as therapy against Acinetobacter (some of them are not active due to intrinsic resistance). The Authors should refer to the list of antimicrobials present in CLSI or EUCAST clinical breakpoints.

- Figure 3. Colistin and polymixin B are not aminoglycosides.

- Antimicrobial activity should be tested according to ISO 20776-1:2019 guidelines using a cation adjusted Mueller Hinton broth with standard inoculum and a proper range of two-fold dilutions for each antimicrobial to determine the minimal inhibitory concentration (MIC, in mg/L). 

Author Response

  1. It is not clear the choice of antimicrobials. Most of the drugs are not used as therapy against Acinetobacter (some of them are not active due to intrinsic resistance). The Authors should refer to the list of antimicrobials present in CLSI or EUCAST clinical breakpoints.

We agree that our original data did not revolve around the antibiotics commonly used to treat Acinetobacter. We have reorganized our data in terms of treatment strategy and included representatives of the major classes in an antibiotic diffusion disc method in our new Table 1.  

  1. Figure 3. Colistin and polymixin B are not aminoglycosides.

We apologize for the oversight. We have now removed these two antibiotics from the aminoglycoside graph and placed them in a miscellaneous antibiotic group in Figure 3C.

  1. Antimicrobial activity should be tested according to ISO 20776-1:2019 guidelines using a cation adjusted Mueller Hinton broth with standard inoculum and a proper range of two-fold dilutions for each antimicrobial to determine the minimal inhibitory concentration (MIC, in mg/L).

We recognize that MICs are the gold standard and we have ordered several antibiotics to perform these tests. Unfortunately, many of the antibiotics are on back-order and will not arrive in the time frame requested by the journal for revisions. In lieu of this data, we have included diffusion disc assays in Muller Hinton Agar according to the standard method (see Table 1). Additionally, we have included this shortcoming as a limitation of this study in the discussion.

Reviewer 3 Report

Acinetobacter baumanii is a nosocomial pathogen commonly known to develop resistance to a number of antibiotics such as beta-lactams, carbapenems, aminoglycosides and fluoroquinolones. In this paper, authors discuss the antibiotic resistance profiles of A. calacoaceticus, a pathogen found along with A. baumanii as a complex. The authors test a wide range of antibiotics against there collection of isolates and share their findings in terms of % growth and compare the resistance profiles of both the species. 

Some of the major concerns are:

1.  The paper is titled " profiling of antibiotic resistance" but there is no MIC data included in the study. The data is based on percentage growth calculations from Phenotype Microarrays. By providing MIC data atleast for clinically relevant resistance and sensitivity differences between both isolates would be helpful.

2. A. calacoaceticus occurs as a complex with A. baumanii, is it possible to discuss this in the introduction and significance of this would help understand the possbility of using antibiotics to which both the strains are sensitive. 

3. Carbapenem is an important class of antibiotics used for Acinetobacter treatment so this needs to be tested and contrasted in both the strains. 

4. Line 66-67 discusses the observation of "little to no expression" from genomic analysis data which is fundamentally not possible from DNA sequence analysis. 

5. Some of the antibiotics tested are irrelevant to gram negative bacteria treatment or Acenitobacter itself and hence it would be good to include classes of antibiotics used to treat these strains, if possible. 

6. Colistin and polymyxin are not aminoglycosides as classified in this study.

7. Is it possible to correlate the MIC data of the isolates with the % growth data from the Phenotype Microarrays? 

Author Response

  1. The paper is titled " profiling of antibiotic resistance" but there is no MIC data included in the study. The data is based on percentage growth calculations from Phenotype Microarrays. By providing MIC data at least for clinically relevant resistance and sensitivity differences between both isolates would be helpful.

We recognize that MICs are the gold standard and we have ordered several antibiotics to perform these tests. Unfortunately, many of the antibiotics are on back-order and will not arrive in the time frame requested by the journal for revisions. In lieu of this data, we have included diffusion disc assays in Muller Hinton Agar according to the standard method (see Table 1). Additionally, we have included this shortcoming as a limitation of this study in the discussion.

  1. A. calacoaceticus occurs as a complex with A. baumanii, is it possible to discuss this in the introduction and significance of this would help understand the possibility of using antibiotics to which both the strains are sensitive.

We agree that this is an important point and we have now included this information in the introduction.

  1. Carbapenem is an important class of antibiotics used for Acinetobacter treatment so this needs to be tested and contrasted in both the strains.

We appreciate the suggestion. We have now included Imipenem and Meropenem in Table 1.

  1. Line 66-67 discusses the observation of "little to no expression" from genomic analysis data which is fundamentally not possible from DNA sequence analysis.

Thank you for identifying our mistake. We have now corrected the statement.

  1. Some of the antibiotics tested are irrelevant to gram negative bacteria treatment or Acenitobacter itself and hence it would be good to include classes of antibiotics used to treat these strains, if possible.

We agree that our original data did not revolve around the antibiotics commonly used to treat Acinetobacter. We have organized our data in terms of treatment strategy and included representatives of the major classes in an antibiotic diffusion disc method in our new Table 1.  

  1. Colistin and polymyxin are not aminoglycosides as classified in this study.

We apologize for the oversight. We have now removed these two antibiotics from the aminoglycoside graph.

  1. Is it possible to correlate the MIC data of the isolates with the % growth data from the Phenotype Microarrays?

Unfortunately, Biolog has not disclosed the exact concentrations used in their plates and it is unlikely that the range is wide enough to allow us to calculate MIC values. To address this issue, we have now added Table 1, which depicts the Kirby Bauer Disk Diffusion Susceptibility Test.

Round 2

Reviewer 2 Report

The Authors correctly addressed all concerns made by the Reviewer. Please add the category interpretation to table 1 to improve clarity according to CLSI clinical breakpoints.

Author Response

Thank you for the suggestion. We have now added the categories to Table 1 . 

Reviewer 3 Report

The authors have answered my comments and suggestions. 

Author Response

We thank the reviewer for their comments and suggestions.